# Setting a Plausible Route for Saline Soil-Based Crop Cultivations by Application of Beneficial Halophyte-Associated Bacteria: A Review

**DOI:** 10.3390/microorganisms10030657

**Published:** 2022-03-19

**Authors:** Han Meng Teo, Aziz A., Wahizatul A. A., Kesaven Bhubalan, Siti Nordahliawate M. S., Muhamad Syazlie C. I., Lee Chuen Ng

**Affiliations:** 1Laboratory of Pest, Disease and Microbial Biotechnology (LAPDiM), Faculty of Fisheries and Food Science (FFFS), Universiti Malaysia Terengganu, Kuala Nerus 21030, Terengganu, Malaysia; hanmeng1995@gmail.com (H.M.T.); dahliasidique@umt.edu.my (S.N.M.S.); syazliebrahim@gmail.com (M.S.C.I.); 2Biological Security and Sustainability Research Group, Faculty of Science and Marine Environment, Universiti Malaysia Terengganu, Kuala Nerus 21030, Terengganu, Malaysia; aaziz@umt.edu.my; 3Institute of Marine Biotechnology, Faculty of Science and Marine Environment, Universiti Malaysia Terengganu, Kuala Nerus 21030, Terengganu, Malaysia; wahizatul@umt.edu.my (W.A.A.); kesaven@umt.edu.my (K.B.)

**Keywords:** salinity issues, halophytes, HT-PGPB, crop improvements, rice

## Abstract

The global scale of land salinization has always been a considerable concern for human livelihoods, mainly regarding the food-producing agricultural industries. The latest update suggested that the perpetual salinity problem claimed up to 900 million hectares of agricultural land worldwide, inducing salinity stress among salt-sensitive crops and ultimately reducing productivity and yield. Moreover, with the constant growth of the human population, sustainable solutions are vital to ensure food security and social welfare. Despite that, the current method of crop augmentations via selective breeding and genetic engineering only resulted in mild success. Therefore, using the biological approach of halotolerant plant growth-promoting bacteria (HT-PGPB) as bio-inoculants provides a promising crop enhancement strategy. HT-PGPB has been proven capable of forming a symbiotic relationship with the host plant by instilling induced salinity tolerance (IST) and multiple plant growth-promoting traits (PGP). Nevertheless, the mechanisms and prospects of HT-PGPB application of glycophytic rice crops remains incomprehensively reported. Thus, this review describes a plausible strategy of halophyte-associated HT-PGPB as the future catalyst for rice crop production in salt-dominated land and aims to meet the global Sustainable Development Goals (SDGs) of zero hunger.

## 1. Introduction

Generally, soil salinization occurs by the localised increment of soluble salt in the surface layer of the soil, which leads to increased soil electrical conductivity (exceeding 4 dS/m) and salinity level. It also implies soils where natural leaching is no longer sufficient to remove the excess salts from the soil profile [1]. The process of land salinization can be classified into primary and secondary salinization. Primary salinization is due to natural causes and mainly originates from two sources, the weathering mineral rocks of the lithosphere and salts from seawater [2,3]. On the other hand, anthropogenic activities such as agricultural use and rapid urbanisation are responsible for secondary salinization [4].

According to the recent data from the Global Map of Salt-affected Soils (GSASmap) of the Food and Agriculture Organization of the United Nations (FAO), information from 118 countries that covers 85% of the global land area showed that more than 423 million ha (3%) of topsoil and 833 million ha (6%) of subsoil are salinized [5]. Hayat and others stated that worldwide over 20% of cultivated lands as well as 33% of irrigated agricultural lands are salinized [6]. Along with this, more than 77 million ha of the salinity-affected lands are human-induced, and 70% of the human-induced salinization is exclusively in Asian regions [7]. Moreover, it is claimed that the soil salinization rate is increasing up to 10% annually due to various factors such as global warming, malpractice in agricultural management, and inevitable natural processes [8].

The varying degree of soil salinity issues significantly impacts the agricultural industries by reducing the economic returns of cultivated land, leading the land to be barren and ultimately to mass land abandonment problems [9]. Hence, saline soil-based cultivation may be an alternate solution to mitigate such issues and cope with the ever-expanding salinity problems, and salt-tolerant crops are the key to this strategy. Previous development trials on salinity and water-deficit stress-tolerant crops were solely through selective breeding and high-tech genetic engineering [10]. Nonetheless, the progress of successful outcomes is mediocre, as proven by the limited number of effective salinity-tolerant genotypes made available so far [11].

Therefore, the biotechnological approach of utilizing beneficial bacteria to improve soil health is vital to ensure global food sustainability. In the natural environment, the interactions of intracellular and extracellular microorganisms are crucial in sustaining and promoting plant growth. Accordingly, multiple types of research and experiments have been undertaken over the years on such microbes, which have proven that these beneficial microbes do indeed qualify to act as bio-fertilizers, bio-stimulants, as well as bio-pesticides. Hence, with the proven evidence on roots and rhizospheric soil of various halophytes harbouring different genera of HT-PGPB, this paper is set to review the plausible growth-promoting potential and mechanisms of HT-PGPB from coastal halophytes in Malaysia in order to induce salt tolerance and augment crop performance of local rice cultivars cultivated in saline affected conditions [12,13].

## 2. Salinization Effects on Plants

A salinized soil is defined as soil with electrical conductivity (EC) value of or exceeding 4 dS/m (40 mM sodium chloride, NaCl) in the root zone at 25 °C, with 15% exchangeable sodium, and a pH value less than 8.5 [8,14]. Various soluble salt ions contribute to soil salinization, and the varying concentration of these soluble salts depends on soil traits and the salinization process [4,15,16]. Despite that, sodium ion (Na^+^) and chloride (Cl^−^) are the most prominent and showed the most phytotoxicities [8,17]. Another variant of soil salinity is soil alkalinity. It occurs when the soil is primarily saturated in sodium carbonate, causing the soil pH to rise, and its effects are more devastating than normal salinized soils [18].

Salinized soils are notoriously known to restrain salt intolerant plant growth worldwide by inducing salinity stress. This abiotic factor significantly affects almost every agricultural development aspect, including lowered productivity and yield potential, disrupting local ecological balance, impairing economic returns, and the precursor of soil degradations and erosions [19]. Typically, the adverse effects of salinity stress are depicted as two consecutive stages. Firstly, the excessive presence of salt reservoirs in the root zones will disrupt the regular water uptakes due to the altered osmotic pressure. This is known as the water-deficit effects of salinity stress, and the symptoms shown include stunted growth and developments due to reduced cell division and differentiation. Moreover, cell deaths will occur due to osmotic stress if there are excessive accumulations of salt ions in the cell walls [20].

Subsequently, ion-specific toxicity occurs when an extreme amount of salt accumulates in the transpiration stream, exceeding the exclusion rate of the plant. This will result in ionic imbalances, disturbance in ion homeostasis, and disrupted water status. During this stage, the stressed plants will decrease productivity and eventually lead to plant mortality. On top of that, oxidative stress will occur as the secondary effect of salinity stress due to the induced reactive oxygen species (ROS) production via photosynthetic imbalance imposed by both water-deficit stress and ion-specific toxicity [21].

Typically, most crops cultivated are glycophytic, meaning that they are able to sustain average growth in somewhat saline soil and will inevitably decline with increasing salinity levels [20]. Such salinity-intolerance plants will undergo various physiological and morphological changes according to the stages of salinity stress, and plant deaths will occur under prolonged or severe stress [22]. Similarly, the high salinity level will also influence the soil’s physicochemical and microbiological properties.

Rice, *Oryza sativa* L., is a cereal crop deemed a staple food among more than half of the world population, and people in Asia have accounted for at least 2 billion consumers [23]. Nonetheless, rice is relatively salt-sensitive that productivity and yield potential will start to decline at 2 dS/m soil salinity and be further deteriorate at 4 dS/m [24,25]. The adverse effects of salinity stress on rice plants are portrayed as symptoms of the diversely affected plant physiologies, such as reduced germination rate [26], retarded seedling and vegetative developments [20,27], leaf mortalities [28], cause panicle sterility [29], and diminished final yield components [30].

In fact, Amirjani has proven the significant effect on rice crops under different levels of salinity stress. The results include decreased plant growth and relative water content, K/Na ratio, sugar content, chlorophyll pigment content, and total protein content. Additionally, the salinity-stressed rice crops indicate increasing proline accumulations due to heightened compatible solute accumulation, and malondialdehyde (MDA) content due to lipid peroxidation [31]. Moreover, rice crops’ salt sensitivity makes up to 70% of yield loss solely due to salinity stress [32]. In addition, with the ongoing land salinization issue, the reduction of arable land globally exacerbates the situation by contributing to the stagnation of rice crop productions and other crucial crops [33].

## 3. Salinity Alleviation Attempts

To date, efforts such as soil reclamations and crop modifications have been carried out in order to mitigate salinization problems and the dire need for fertile agricultural lands [34]. The standard methods on soil reclamations comprise of physical means (such as breaking down and disturbing the soil surface layer, scraping away the soil surface salt build-ups, and flooding and leaching), chemical amendments (including gypsum, calcium chloride, and lime), and some agricultural managements such as proper soil amendments, irrigation methods, doing crop rotations, intercropping, or precision farming [20,35,36,37] However, such implementations, in reality, are always constrained by limited costs and water source quality [37].

Apart from soil reclamations, crop modifications via traditional selective breeding, primming agent applications, genetic engineering, and induced mutant breeding were also carried out by researchers aiming to augment crops with new potential qualities, such as heightened salt tolerance [38,39,40]. However, it has proven to be a difficult task as salinity stress is complex. It affects multiple facets of plant physiology, not to mention the time, effort, and costs needed [41].

Ironically, to counteract the low productivity and yield of salt-sensitive crops such as rice on salinized soils, farmers usually opted to apply chemical fertilizer to mitigate the problems because of convenience. Furthermore, due to the vulnerability of salinity-stressed plants to pests and disease, excessive usage of pesticides is usually involved. These temporary solutions will eventually lead to even more build-ups of soluble salts from the chemical amendments, hastening or worsening the salinity problems [42,43] Moreover, some pesticides such as parathion can attain further stability and become more structurally non-degradable under saline conditions, causing long-term soil contamination [44].

Hence, the biotechnological approach of bio-inoculants, namely HT-PGPB, is demonstrated to be an effective alternate measure of mitigating salinity stress effects and overall crop improvements [45]. Naturally, these beneficial microbes possess considerable growth-promoting and salinity-defensive traits that improve the inoculated host’s tolerance towards salinity and even biotic stresses [46]. The HT-PGPB alleviates salinity stress through various mechanisms evoking multipronged physiological, biochemical, and molecular plant responses to promote plant growth, as shown in Figure 1.

## 4. Halophytes and Halotolerant Plant Growth-Promoting Bacteria (HT-PGPB)

The salinity tolerance revolves around a complex of physiological, morphological, and molecular processes, alongside the determined factors of intensity and duration of exposures. To illustrate, the capability of a plant to limit toxic salt intake and accumulations, regulate cells’ ionic and osmotic balance, and control leaf development and the onset of senescence are examples of salinity-tolerant mechanisms [47]. For instance, halophytes are plants that thrive and complete their life cycle in an environment with elevated salinity (up to 1 M sodium chloride, NaCl) without suffering from any sign of salinity distress [43,48]. Additionally, some coastal region halophytes can even reach optimum growth at 5 to 25% salinity level of standard seawater. Nevertheless, their growths can still be affected by either absence or over accumulations of salt [49].

Consequently, halophytes are equipped with diverse natural strategies to combat the adverse effects of excessive salt accumulations. The halophyte’s primary mechanisms are involved in their morphological (thickened leaves, foliar salt glands), physiological (salt elimination mechanisms), biochemical (accumulation or synthesizing compatible solutes), and salt-responsive genet characteristics such as senescence-associated genes (SAG), homeostasis genes (SOS), molecular chaperones (HSP), and dehydration-related transcription factors (DREB) [50].

The secondary mechanism alternately is associated with the beneficial microorganisms present in the plant’s rhizosphere, endosphere, and phyllosphere [50]. These microorganisms comprise non-pathogenic, halophilic, or halotolerant bacteria or fungi with plant growth-augmenting factors. Initially, these microorganisms are driven by the secreted root exudate that originates from photosynthetic activities. They utilize it as nourishment and to produce advantageous secondary metabolites that aid the host plants. For example, L-tryptophan secreted among root exudates is metabolized into the phytohormone IAA that aids in promoting root growth. The organic acid byproducts of the bacteria can aid in lower soil pH to encourage nutrient mobility.

The prospects of plant growth-promoting bacteria (PGPB) applications are well established in improving plant health and diminishing the effects of biotic and abiotic stresses, all without any significant or adverse effects on well-growing plants under optimal soil conditions [51,52,53]. Correspondingly, HT-PGPB derived from various halophytes or saline soils are ventured among researchers as bio-inoculants in aiming to enhance glycophytic salt tolerance, primarily on salt-sensitive crucial crops such as maize, wheat, and rice [54]. A halotolerant microorganism is described by its survivability and tolerance in an environment or media with a wide range of NaCl concentrations: it can be as high as 25%, optimal at 1% (<0.2 M), or in the absence of NaCl at all [55,56]. Hence, utilizing HT-PGPB isolated from the saline environment is crucial as particular specific genetics, and physiological characteristics allow them to thrive and sustain the often-extreme conditions of the targeted field [57]. Moreover, Khan et al., discovered that some plant growth-promoting rhizobacteria (PGPR) had a decreased efficacy due to the contrast of environmental factors of the targeted field from the environment the PGPR originates [58].

Table 1 depicts the different types of HT-PGPB isolated from diverse halophytes with proven multiple beneficial traits that can act on inoculated glycophytic crops. Likewise, Hassan et al. utilized grounded dried root powder of multiple coastal halophytes as carriers of localized endophytic HT-PGPB and reported success in improving wheat growth under salinized conditions [59].

These plant–microbial interactions are incredibly complex and dynamic. Generally, the action mechanisms of the HT-PGPB enable them to be categorized as bio-fertilizer that improve soil nutrient bio-availabilities, bio-stimulators that synthesise exogenously, or stimulate the production of endogenous phytohormones, and as biopesticides that deploy antagonistic effects on invading plant pathogens [70]. Besides, the different mechanisms can also be grouped into direct mechanisms that consist of plant growth-promoting and salinity defense factors. The direct mechanisms incite the plants’ metabolism that collectively led to their saline adaptive augmentation, also known as induced salinity tolerance (IST) [71]. The anti-phytopathogenic protective mechanism of the HT-PGPB falls under the indirect mechanism. This group of mechanisms comprises countermeasures to protect the host plant from pathogens via secretion of antipathogenic substances such as antibiotics and volatile organic compounds, exert competitions for various needs, and instigate induced systemic resistance (ISR) within the host plant [72,73].

Progress in molecular and plant biotechnology revealed that HT-PGPB alleviates salinity stress through a complex network of signally activities during plant-microbial interactions. These synergistic interactions start with adaptations of the newly introduced HT-PGPB to the often-hyperosmotic rhizosphere, followed by instilling multiple salt-tolerance events to the host plant, and lastly, deploying improving mechanisms on the surrounding soil quality [74].

## 5. Plant Growth-Promoting Mechanisms by HT-PGPB

### 5.1. HT-PGPB Mediated Soil Nutrient Bio-Availabilities

Deficiencies of plant-available soil nutrients are critical, especially for plants under salinity stress. The physicochemical properties of salinized soil reduces various nutrient bioavailability, and thus symptoms of deficiencies are common among salinity-stressed plants [75,76,77]. To prevent such problems, temporary solutions of continuous and unregulated applications of chemical fertilizer are practiced, which will lead to environmental hazards, soil health deterioration and, ironically, further increase the soil salinity level [78]. On that account, the applications of beneficial microorganisms to increase nutrient bio-availability rather than chemical amendments are stated to be more sustainable and greener solutions for crop production systems. For instance, the capability of HT-PGPB in increasing essential soil nutrient bio-availability is addressed by a wide range of different action mechanisms, as shown in Table 2, which is crucial for promoting plant health and growth.

Nitrogen (N) is one of the macronutrients that plays multiple essential roles in plant growth and productivity, mainly involved in the cellular synthesis of proteins, enzymes, DNA, and RNA. Although there is an abundance of atmospheric N (78% of the atmosphere), plants cannot utilize it in such form. Therefore, bacteria with nitrogen-fixing ability are crucial in metabolizing and conversion of atmospheric N into plant-available ammonium (NH_4_^+^) and nitrate (NO_3_^−^) forms [84]. Moreover, the high Cl^−^ uptake from the saline soil significantly reduced the absorption of nitrogen and sulfur [75]. That being the case, halotolerant diazotrophic PGPB are proven capable of fixing atmospheric N via a loose symbiosis mechanism also known as biological nitrogen-fixing (BNF) to remediate reduced available N source [85,86,87]. These N-fixing bacteria are free-living rhizobacteria that are either mainly ectorhizospheric or endophytes within the plants [88]. Franche and colleagues refer to these diversities of diazotrophic bacteria as associative N-fixing bacteria that do not involve the usual nodule formations [89].

The N-fixing mechanism by the bacteria is due to the possession of *nif* genes which encode for the production of nitrogenase enzymes that reduce atmospheric N into ammonia, and *fix*ABX genes, which are required for the nitrogen fixation process for free-living bacteria [90]. The fixed plant-available N is directly taken up by the host plant, hence remediating N deficiencies by a certain level under salinity stress [91].

Phosphorus (P) is an essential macronutrient that is the requisite element for physiological processes such as photosynthesis, energy transfer, biosynthesis of macromolecules, and respiration [92]. The approximate P content in soil is often 0.05% (*w*/*w*). Only 0.001% of the total P as inorganic forms of primary and secondary ions of orthophosphates (H_2_PO_4_^−^ and HPO_4_^2^^−^) are absorbable by plants. This is due to the poor solubility and tendency to form organic P minerals in the soil in the form of apatites (hydroxyapatite and oxyapatite) that are not readily leached [92]. Consequently, P deficiencies are common for plants suffering from high salt stress [93,94].

Application of HT-PGPB with phosphate-solubilizing capability, also known as phosphate-solubilizing bacteria (PSB), can significantly aid in increasing bio-availabilities of soil P [80]. It is proven that the application of PSB in a field can reduce the required P-type fertilizers usage by approximately 50 % without affecting the final yield [95].

The mechanisms of PSB involve mineralizing organic P (tricalcium phosphate, aluminium phosphate, rock phosphate, etc.) in the soil by solubilizing their complex structure, releasing P as inorganic form. The multiple solubilizing mechanisms that PSB utilizes include chelation, acidification, ion-exchange reactions, and organic acid productions [86]. Primarily, the PSB secretes low molecular organic acids, which are the by-product of metabolized sugars from the root exudates [96]. Examples of these low molecular weight organic acids are gluconic acid, citric acid, succinic acid, propionic acid, and lactic acid [97]. These acids act as chelators of divalent calcium cations (Ca^2+^), thus releasing the insoluble bonded P ions in inorganic forms. The secreted acids also help lower the surrounding soil pH to maintain the mobilisation of available P [98]. Besides, some PSB directly lowers soil pH by directly releasing hydrogen ions. Furthermore, PSB also synthesis phosphatases or phytase enzymes that hydrolyses organic P in the soil [99]. Additionally, HT-PGPB possesses the ability to solubilize organic potassium (K) and zinc (Zn) in the soil similarly via organic acid secretions and altering the surrounding soil pH [16,83,85].

Iron (Fe) is the fourth most abundant element on Earth and one of the essential micronutrients that act as a co-factor of enzymes involving some crucial physiological processes such as photosynthesis, respiration, and nitrogen fixation. Plant-absorbable Fe is soluble ferrous ion (Fe^2+^), which is highly unstable in aerobic conditions and will be oxidized into ferric ion (Fe^3+^). Ferric ion subsequently tends to form insoluble compounds (ferric hydroxides, oxyhydroxides, and oxides), resulting in the meagre amount of Fe available for microbial and plant assimilation [100,101].

To alleviate the limitations of Fe supply, HT-PGPB can increase soil Fe bio-availability by synthesizing siderophores, a low-molecular compound (0.5–1 kDa) with functional groups of hydroximates and catechols which have a very high affinity for ferric ions and a reversible way of binding [102,103]. The formed bacterial siderophore-Fe complexes can be easily accessible and taken up by plants through the destruction of chelate or direct absorptions [104]. The other theory that has been suggested is that this occurs through ligand exchanges. The plant-produced siderophores (phyto-siderophores) interact with the bacterial siderophores-Fe complex, initiating the ligand exchange reaction, and the plants absorb Fe via the iron transferred phyto-siderophores [105].

### 5.2. HT-PGPB Mediated Indole-3-Acetic Acid (IAA) Production

Generation of phytohormones is not exclusive to plants as plant-accompanied HT-PGPB possess the ability to alter plant phytohormone levels by producing exogenously [106]. Under normal circumstances, IAA is involved in cellular division and enlargement, translating into seed germination, shoot growth, and root initiation. However, plants will generally suffer from a drastic drop in IAA concentrations under salinity stress. For example, IAA cumulation of tomato is affected at 100 mM NaCl and further diminishes at 300 mM and above [107]. Consequently, such a decline will reduce germination rate, retarded root formations, and stunt plant growth and development [108]. Rice plants are no exceptions, as the stated symptoms are also shown on studied rice seedlings under simulated saline conditions.

Therefore, the application of HT-PGPB with IAA synthesizing ability can mitigate the low cumulation by producing exogenously to be taken up by the host plant [109]. The production of exogenous bacterial IAA involved utilizing L-tryptophan (L-Trp amino) secreted among root exudates or decaying cells as precursors [110]. There are currently five L-tryptophan-dependent pathways documented. The biosynthesis process is subject to root exudate contents and environmental factors such as soil salinity level, pH, and osmotic or matrix stress [111].

The synthesized exogenous IAA aid in balancing IAA levels in the roots, stimulating root proliferation, increasing root size and weight, promoting root exudations, and developing lateral roots to achieve more extensive surface contact in the soil. Thus, the improved root system will lead to better ability to probe the soil for more nutrient exchanges, water uptakes, growth capacity, and indirectly help maintain leaf growth to retain photosynthesis rate and subsequently productivity of the plant even in the high salt environment [107,112,113].

As shown in Table 3, the isolated IAA-producing microbes significantly enhanced plant growth in biomass under the simulated saline condition in terms of shoot length and weight (fresh and dry), emphasizing root length and weight (fresh and dry), and germination rates of salt-sensitive crop seeds. Furthermore, the selected microbes will have a slight improvement to no significant effects on non-stressed control plants [63,104,114,115].

## 6. Salinity Mitigating Mechanisms by HT-PGPB

### 6.1. HT-PGPB Modulations of Stress Ethylene

Ethylene is one of the stress-signalling phytohormones that, at a low amount, initiates various response mechanisms to counteract biotic or abiotic stress effects, including salinity stress [70]. However, prolonged and severe stress will lead to the second peak synthesis of ethylene, also known as excessive or stress ethylene. The deleterious effect of stress ethylene includes stunted root development, which subsequently impairs root functioning, reduces vegetative growths, and eventually affects productivity and yields [94,119].

Ethylene synthesis in plants begins with its precursor, 1-amino-cyclopropane-1-carboxylic acid (ACC) that is converted by the enzyme ACC oxidase to the final product of ethylene. Due to that, some HT-PGPB with the possession of acdS genes can metabolise ACC, which is secreted among root exudates into ammonia and α-ketobutyrate via the production of enzyme ACC deaminase (ACC-D). These bacteria then utilize the end products as their unique C and N source [120]. Hence, HT-PGPB with ACC-D activity of at least 20 nmol α-KB mg^−1^ h^−1^ can significantly reduce the total pool level of ethylene precursors of stressed plants and is said to potentially lower the second peak of ethylene production by 50% to 90% [121]. After that, the reduced ethylene level will enhance the plant’s stress tolerance level, thus promoting plant growth and development even under unfavourable conditions [120].

The molecular analysis documented that HT-PGPB such as *Bacillus pumilus*, *Zhihengliuella halotolerans* [116], *Variovorax paradoxus*, *Arthrobacter agilis* [81], and *Micrococcus yunnanensis* [64] possess ACC deaminase activities that significantly increase the inoculated plant’s salt tolerance and growth under salinized conditions.

### 6.2. HT-PGPB Modulations of Exopolysaccharides

Under undesirable conditions, soil bacteria secrete polysaccharides to promote adherence to available environmental surfaces and form an organo-mineral sheath, also known as a biofilm. These polysaccharides consist of complex mixtures of polymers with high molecular weight (MW ≥ 10,000) that provide both physical and functional protection against desiccating conditions and constraints of high salinity [122]. These extracellular polysaccharides are vital components of biofilm formations and effectively alleviate salinity stress [123]. Furthermore, a rhizosheath is a form of EPS that layers around the root surface. The rhizosheath serves as a barrier against toxic ions, site of nutrient cycling, cation uptakes, symbiotic reactions, and maintaining osmotic equilibrium [124,125]. EPS productions are typical for bacteria under heavy metal and high salinity stresses [126]. However, high-quality EPS can only be produced by halo- or drought-tolerant rhizobacteria to tolerate and survive under harsh conditions [127].

EPS produced by HT-PGPB possess multiple functionalities such as enhancing soil structures by crumb formations, increasing macropore volumes, and aggregations of rhizospheric soil, resulting in increased water retention and provision of nutrients for the host plants. More importantly, EPS also possess chelation capabilities, which can monitor cation intake by chelating excess Na^+^ in the rhizosphere, rendering them immobilized and unavailable around the root surface (binding sites include hydroxyl, sulfhydryl, carboxyl, and phosphoryl group), therefore notably reducing the amount of Na^+^ uptakes and preventing ion toxicities [126,128,129]. Ashraf and others discovered that the alleviation of salinity in wheat inoculated with EPS-producing HT-PGPR is due to the reduced Na uptake in roots, and in addition to that, is restricted to be transferred to the leaves [127]. It is reported that EPS producing HT-PGPB includes *Pseudomonas mendocina,* which were inoculated on lettuce [130], and *Halomonas variabilis* and *Planococcus rifietoensi*, which have been proven to improve surrounding soil structures of chickpea [131].

### 6.3. HT-PGPB Modulation of Antioxidant Defences

Oxidative stress is the follow-up secondary effect of salinity stress. The reduced photosynthetic activity caused by salinity stress effects led to the over-reduction of photosynthetic electrons, thus generating reactive oxygen species (ROS) [132,133]. The generated ROS such as hydrogen peroxide (H_2_O_2_), superoxide ions (O^2−^), singlet oxygen (^1^O_2_), and hydroxyl radical (OH^−^) are toxic molecules that are highly reactive [134]. They tend to cause oxidative damage to plant biomolecules such as membranous lipids and proteins and nucleic acids, which results in disrupted metabolic enzyme activities and cell homeostasis [123,135,136]. Kim et al. reported that membrane deterioration due to ROS, which leads to cellular toxicity, has been discovered in salinity-stressed rice, citrus, and tomato plants [137].

The inoculation of HT-PGPB can confer stress tolerance by improving the antioxidant status of the plants [138]. Some HT-PGPB alleviate salinity through synthesizing scavenging enzymes or antioxidants such as enzymatic ones (superoxide dismutase, catalase, ascorbate peroxidase, polyphenol oxidase, and glutathione reductase) and non-enzymatic types (ascorbate, glutathione, carotenoids, and phenolics) to assist in degrading ROS into harmless compounds or instigate antioxidant gene expression of the host plant [123,139].

For instance, the introduction of HT-PGPB with antioxidative properties enhanced soybeans’ gene expression of APX, CAT, and SOD when exposed to saline stress [140]. HT-PGPB with antioxidative activities reported also includes increased catalase and glutathione activities by *Bacillus cereus*, *Bacillus subtilis*, and *Pantoea ananatis* obtained from roots of *Rhizophora apiculata* [141], and *Bacillus pumilus* with catalase, glutathione, and polyphenol oxidase activities isolated from *Avicenna marina* [118].

### 6.4. HT-PGPB Modulations of Osmotic Balance Regulation

Homeostasis of ion concentration is crucial in plant cells under salinity stress. The excessive uptake of toxic salt ions like Na^+^ and Cl^−^ can upset the balances of other vital ion intakes, namely vital potassium ion (K^+^) accumulations. Choudhary stated that the chemical physiological properties of K^+^ are highly similar to Na^+^, meaning that Na^+^ can compete with K^+^ binding sites of various crucial enzymatic reactions, protein synthesis, and ribosome functions. By improving the plant’s selective uptake of K^+^, the accumulation of toxic Na^+^ ions diminish significantly, thereby maintaining an optimal high K^+^/Na^+^ ratio [94]. The high K^+^/Na^+^ ratio is able to suppress the osmotic stress of the plant by preserving higher stomatal conductance as well as photosynthetic processes [107].

HT-PGPB applications were proven to assist mitigate salinity stress by alleviating toxicity of Na^+^ by enhancing selective uptakes of nutrients within the plant cells [142]. These HT-PGPB can constrict Na^+^ uptakes by modifying the plant cells’ cell wall or membrane composition. Modifications include upregulating the NHX (cation/proton antiporter) and HKT (high-affinity potassium transporter) transporter, which is responsible for selective uptake of K^+^ and translocation of toxic Na^+^ ions, increasing the electrogenic Na^+^/H^+^ ionic-porters, along with enhanced expression of salt overly sensitive (SOS) genes [143,144].

The accumulations of osmoprotectants, also known as osmolytes or compatible solutes, are small low-toxicity organic compounds synthesized by HT-PGPB that can aid in regulating the plant’s osmotic balance during salinity stress [145]. Plants inoculated with HT-PGPB are proven to have improved water relationships and balanced root hydraulic conductivity through accumulations of osmoprotectants [97]. Examples of osmoprotectants include soluble sugar derivatives (sucrose, trehalose, maltose, cellobiose, and turanose), amino acids (glutamate, proline, alanine, serine, threonine, and aspartic acid), quaternary amines (glycine, betaine, and carnitine), imino acids (pipecolate), and tetrahydropyrimidines (ectoines) [146,147,148].

The osmoprotectants help stabilize protein from denaturing during high salt concentrations, maintaining cytosolic pH, stabilizing membrane integrity, reducing cell osmotic potential, maintaining turgor pressure, and balancing cell redox condition [149,150,151]. Additionally, one major factor of glycophytes’ poor salt tolerance is its insufficient accumulation of compatible solutes, and stressed plants prefer the uptake of microbe-liberated osmoprotectants rather than synthesizing de novo themselves to conserve energy under stress conditions [152]. *Azospirillum* spp., *Pseudomonas* spp., *Bacillus* spp., and *Rhizobium* spp. are examples of HT-PGPB that are able to produce a high quantity of osmolytes in saline habitats [145].

Generally, HT-PGPB-inoculated crops yield superior outcomes in terms of enhanced salinity tolerance. The inoculated glycophytes will have a better relative water content, higher or reduced proline accumulations (that varies among various researches due to the different nature of the HT-PGPB and the status of the inoculated plant), optimal K^+^/Na^+^ ratio, higher chlorophyll pigments content, and lowered lipid peroxidation activities [118,141,153,154].

## 7. Future Developments and Challenges

Overall, the plant growth-promoting traits and salinity defence conferred by inoculated HT-PGPB have been proven capable of instilling IST among targeted glycophytic crops. In some cases, the isolated HT-PGPB has been proven to be non-specific to the original host or source.

Notwithstanding this, there are observed cases when a control HT-PGPB has no significant effect on other crops, showing that certain factors such as rhizospheric environment, root exudate content, or root morphology need to be fulfilled before successful colonization, and thus inoculation is achievable [155]. Kamilova and others discovered that the disparity in the efficacy of an HT-PGPB could also be due to the multiplicity of climatic and environmental factors that vary from one farm to another or even within fields [156].

Rice in Malaysia is the third most important crop but has yet to achieve a self-sufficient level due to stagnation-causing factors. Salinity stress is one of the problems faced, as Malaysia is not spared soil salinization problems. Rising sea levels and saline water intrusions threaten the major granaries along the coastal region [157]. It is calculated that approximately over 180,000 ha of agricultural land in Malaysia will be lost with each metre rise in sea level, and the salinity problem will affect an area of rice cultivation of up to 1,000,000 ha by the year 2056 [157,158]. Moreover, with the population in Malaysia estimated to reach 66.4 million by the year 2056, advancements in rice production are needed to meet the consumption requirements within the nation [158]. Malaysia is home to diverse native halophytes species, which harbours potential HT-PGPB. It is postulated that some of these halophyte-associated HT-PGPB may possess the desired traits (PGP and salinity defence) that can be host-compatible with the local rice cultivar.

However, information regarding the plant–microbial interactions between native isolated HT-PGPB with the local rice cultivar is scant, as only a handful of research works have been published over the years. For example, Deivanai et al. isolated various species of endophytic *Bacillus* spp. and *Pantoea* sp. from petioles of *Rhizophora apiculate* that enhance the growth of rice seedlings, and Shultana et al. isolated rhizobacteria *Bacillus tequilensis* from a local salinity-affected rice field that improves the growth of salt-sensitive rice cultivars [67,141].

Hence, the discoveries of more rice plant-compatible HT-PGPB from native halophyte plants as bio-inoculants can potentially provide breakthroughs in future sustainable agriculture prospects and mitigate the ongoing salinization issues. Lastly, the guideline states that an ideal PGPB should be an aggressive colonizer, possess multiple PGP traits with non-host specificity, not be antagonistic to local microbes, be isolated from indigenous salt-affected soils, and be compatible with inoculant carriers [13]. Moreover, the PGPB should not benefit nearby wild or invasive plants and be stabilized enough to not further genetically evolve with undesirable traits [159]. Thus, the success of HT-PGPB helps to determine an alternative strategy of saline soil-based rice cultivation, aiming to preserve future food security.

## 8. Conclusions

Soil salinity has become a menace to soil that poses a significant threat to worldwide agricultural industries, and the go-to unsustainable chemical treatments exacerbate the situation. Moreover, Malaysia’s rice-producing sectors are at a stagnant level, with land salinization as part of the problem while waiting for any scientific breakthroughs. Thus, the exploration of HT-PGPB as bio-inoculants would make significant advancements toward sustainable agriculture. The benefits of HT-PGPB have gained great interest in past decades, with multiple studies having demonstrated the capabilities, mechanisms involved, and potentials of HT-PGPB as an optimal and eco-friendly alternative to remediate salinity stress and growth enhancements among salt-sensitive crops. Even so, further in-depth research is required to elucidate and illustrate the varying plant–microbe interactions under complex stresses elicited by soil salinity. A better understanding could set the plausible prospects of advanced saline soil-based cultivation, developing a novel market-available HT-PGPB-based biofertilizer, or even genetically engineering HT-PGPB to obtain the ideal strains.

## Figures and Tables

**Figure 1 microorganisms-10-00657-f001:**
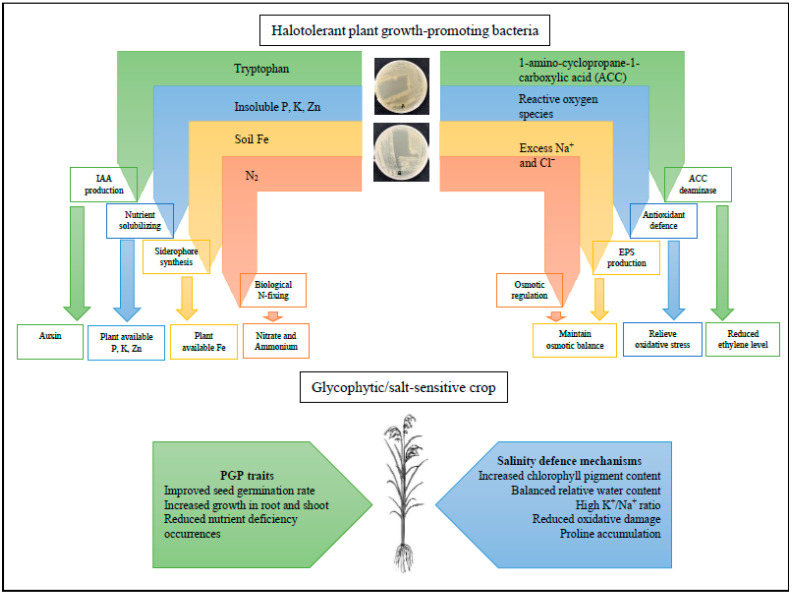
The roles of potential halotolerant plant growth-promoting bacteria (HT-PGPB) with plant growth (PGP) and salinity defense mechanisms to alleviate salinity stress in glycophytic crops. The depicted plates are potential HT-PGPB (rhizobacteria and endophytes, respectively) isolated from native halophytes.

**Table 1 microorganisms-10-00657-t001:** The plant growth-promoting (PGP) traits of isolated HT-PGPB with their proven action mechanism on the inoculated crops.

HT-PGPB	Source	Crop	PGP Traits	Citations
*Micrococcus yunnanensis*	*Avicennia marina*	Rice (*Oryza sativa*)	IAA ^1^, Siderophore ^2^, Ammonia ^3^	Soldan et al., 2019 [60]
*Arthrobacter pascens*	*Suaeda fruticosa*	Maize (*Zea mays*)	P-solubilization ^4^, Siderophore, Antioxidants ^5^	Ullah and Bano, 2015 [61]
*Bacillus* sp.	*Atriplex leucoclada*	Maize (*Zea mays*)	P-solubilization, Siderophore, Antioxidants	Ullah and Bano, 2015 [61]
*Staphylococcus equorum*	*Salicornia hispanica*	Tomato (*Solanum lycopersicum*)	ACC deaminase ^6^, P-solubilization, BNF ^7^, Siderophore	Vega et al., 2019 [62]
*Bacillus atrophaeus*	*Suaeda mollis*	Wheat (*Triticum aestivum*)	P-solubilization, IAA, BNF	Kerbab et al., 2021 [63]
*Arthrobacter agilis*	*Halocnemum strobilaceum*	Sugar beets (*Beta vulgaris*)	P-solubilization, IAA, ACC deaminase	Zhou et al., 2017 [64]
*Alcaligenes faecalis*	*Atriplex lentiformis*	Wheat (*Triticum aestivum*)	P-solubilization, IAA, N-fixing, Antioxidants, Ammonia	Muhammad et al., 2021 [65]
*Pantoea ananatis*	*Oryza sativa*	Rice (*Oryza sativa*)	ACC deaminase, P-solubilization, IAA, Siderophore	Lu et al., 2021 [66]
*Bacillus tequilensis*	*Oryza sativa*	Rice (*Oryza sativa*)	IAA, N-fixing, P-solubilization, K-solubilization ^8^, EPS ^9^	Shultana et al., 2020 [67]
*Bacillus aryabhattai* and *Arthrobacter woluwensis*	*Coastal plants*	Soybean (*Glycine max*)	IAA, EPS, Antioxidants	Khan et al., 2021 [68]
*Bacillus velezensis*	NA *	Tomato (*Solanum lycopersicum*)	IAA, P-solubilization, BNF, Antioxidants	Medeiros and Bettiol, 2021 [69]

^1^ Indole-3-acetic acid synthesis, ^2^ siderophore production, ^3^ ammonia production, ^4^ phosphorus solubilization, ^5^ antioxidant activities, ^6^ 1-aminocyclopropane-1-carboxylic acid (ACC) deaminase activity, ^7^ biological nitrogen-fixing, ^8^ potassium solubilization, ^9^ exopolysaccharides production. * Information not available.

**Table 2 microorganisms-10-00657-t002:** Halophytes associated HT-PGPB with the ability to improve different soil nutrient bio-availabilities.

HT-PGPB	Halophyte	Nutrients	Mechanisms	Citation
*Bacillus atrophaeus*	*Suaeda mollis*	Nitrogen	BNF	Kerbab et al., 2021 [63]
*Alcaligenes faecalis*	*Atriplex lentiformis*	Nitrogen	BNF	Muhammad et al., 2021 [65]
*Agrobacierium tumefacien*	*Arthrocnemum indicum*	Nitrogen	BNF	Sharma et al., 2016 [79]
*Variovorax paradoxus*	*Suaeda physophora*	Iron	Siderophore	Zhou et al., 2017 [64]
*Micrococcus yunnanensis*	*Nitraria tangutorum*	Iron	Siderophore	Zhou et al., 2017 [64]
*Alcaligenes faecalis*	*Sesbania aculeata*	Phosphorus	P-solubilization	Muhammad et al., 2021 [65]
*Enterobacter asburiae* and *Arthrobacter aurescens*	*Echinochloa stagnina*	Phosphorus	P-solubilization	Safdarian et al., 2018 [80]
*Bacillus endophyticus* and *Bacillus tequilensis*	*Salicornia europaea*	Phosphorus	P-solubilization	Zhao et al., 2016 [81]
*Bacillus mucilaginosus* and *Azotobacter chroococcum*,	NA	Potassium	K-solubilization	Singh et al., 2010 [82]
*Azospirillum lipoferum*	NA	Zinc	Zn-solubilization ^1^	Tariq et al., 2007 [83]

^1^ Zinc solubilization.

**Table 3 microorganisms-10-00657-t003:** Mode of actions of IAA-producing HT-PGPB on inoculated crops.

HT-PGPB	Halophyte	Targeted Crop	Mode of Actions on Plant	Citation
*Variovorax paradoxus*	*Suaeda physophora*	Sugar beet (*Beta vulgaris*)	Enhance the growth of shoot and root	Zhou et al., 2017 [64]
*Planococcus rifietoensis*	*Kalidium capsicum*	Sugar beet (*Beta vulgaris*)	Enhance the growth of shoot and root	Zhou et al., 2017 [64]
*Alcaligenes faecalis*.	*Sesbania aculeata* and *Atriplex lentiformis*	Wheat (*Triticum aestivum*)	Enhance growth parameters and plant biomass	Muhammad et al., 2021 [65]
*Bacillus atrophaeus*	*Salicornia* spp.	Wheat (*Triticum aestivum*)	Enhance the growth of shoot and root	Safdarian et al., 2020 [80]
*Bacillus* sp., *Pseudomonas* sp., and *Microbacterium* sp.	*Limonium sinense*	*L. sinense*	Enhance the growth of shoot and root	Qin et al., 2014 [116]
*Bacillus* sp., *Marinobacterium* sp., and *Sinorhizobium* sp.	*Psoralea corylifolia* L.	Wheat (*Triticum aestivum*)	Enhance germination rate and root elongation	Sorty et al., 2016 [117]
*Bacillus pumilus* and *Exiguobacterium* sp.	*Avicennia marina*	Tomato (*Solanum lycopersicum*)	Enhance the growth of shoot and root, leaf numbers and internodes	Ali et al., 2017 [118]

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
