# Peer review of "Setting a Plausible Route for Saline Soil-Based Crop Cultivations by Application of Beneficial Halophyte-Associated Bacteria: A Review"

_microorganisms, 2022, doi:10.3390/microorganisms10030657_

Round 1

Reviewer 1 Report

Starting from the general, defining aspects of saline stress, the authors describe the “plausible strategy of halophyte-associated HT-PGPB” for rice cultivation, in conditions of saline stress.

In the first 6 chapters, the authors describe the salinization effects on plants, some salinity alleviation attempts, the roles of potential HT-PGPB in salinity defense, salinity tolerant mechanisms of halophytes, the capability of HT-PGPB in increasing essential soil nutrient bio-availability, role of HT-PGPB in IAA production, salinity mitigating mechanism by HT-PGPB. Chapter 7 presents the personal opinions, in connection with some quoted sources, regarding future developments and challenges. The authors refer to the rice crop in Malaysia. Chapter 8 presents the conclusions. This chapter could also include references to rice cultivation in Malaysia.

The work includes a figure that represents an original scheme. The authors also wrote 3 tables, citing numerous bibliographic sources.

As mentioned in the paper, the aim is to increase the productivity of rice crops on saline soils. Perhaps it would be useful to mention in the title the fact that it is about rice cultivation. In this case, a new keyword must be added: rice.

The work is very complex and correctly structured. I would add, at least as a single mention, at some point in the introduction for example, the notion of alkalinity.

Among the measures to counteract the effects of saline stress could be cited works that test various products, extracts, etc., applied to the soil or plants, which improve the level of physiological processes…lines 124 – 129.

Citations must be checked. In lines 43-45, the text does not seem to correspond to the quoted source, number 5. Consulting the source, I did not identify the content expressed in this paragraph.

Also, the content related to the exact data of the affected areas, etc., should cite much more recent sources.

For some statements, the original source should be cited or as close as possible to the original. for example, lines 49-51 cite a source from 2020, but which, in turn, cites, in connection with this, older sources. In addition, the statement is found in older works, being taken almost entirely:

Jamil A., Riaz S., Ashraf M., Foolad M.R. Gene expression profiling of plants under salt stress. Crit. Rev. Plant Sci. 2011;30(5):435–458

Shrivastava, P., & Kumar, R. (2015). Soil salinity: A serious environmental issue and plant growth promoting bacteria as one of the tools for its alleviation. Saudi journal of biological sciences22(2), 123-131.

Author Response

Dear editors, 

I am pleased to inform you that we have carefully revised the manuscript accordingly to the suggestions given by the reviewers. Please find the correction as in the attached file. 

Regards, 

Ng 

Reviewer 2 Report

The paper describe very well the results available to date. I suggest a minor revision of english language and spell check, that in some a few parts it doesn't make a fluent reading. The paper is updated and certainly very interesting. 

Author Response

Dear reviewer, 

We have carefully revised and corrected the whole manuscript using a well-known English grammar checking program and proofread by a native English speaker. The have also listed point by point the corrections been done in the attached file. 

Regards, 

Ng 

Reviewer 3 Report

Dear Authors,

The submitted manuscript presents mostly well-written review about alleviating salinity stress in plants using beneficial bacteria. The review cites recent publications together with publication important in the field and covers the description of bacteria-related mechanisms of improving salt tolerance together with socio-economic aspects. This is a very good and important work and that is why it is worth further improving before the publication. I have several general and specific points to improve listed below.        

Language

The quality of writing is mostly good but not consistently good, I suggest giving the manuscript one more intensive check especially in the last paragraphs.  

General comments

- After reading the manuscript I am still wondering what is the “plausible route” mentioned in the title? Authors should relate to the concept introduced in the title in perspectives/conclusions.

- My feeling is that the review does not focus enough on rice especially in the paragraphs 3-6. Please, highlight the discoveries made in your plant of interest or highlight the potential of using strains/mechanisms from related plants (other monocots). Focusing on rice will make this review different from other in the topic.

- I sometimes see that the authors overgeneralize, for example summarizing some works concerning a specific plant to plants in general (some examples below).    

Specific comments

- L19-20 “productivities..” , do you mean “productivity and yield” ?

- L29-30 – “Global Sustainable Development Goal” – what is this goal? I guess that you mean United Nations Global Sustainable Development Goals , but it is difficult to get it from the context. Please specify what goal do you have in mind.

- Figure – The figure is clear but needs improvement to increase readability and quality

-- change the width of the arrows to uniform

-- the photo of PGPR is low quality and I am not even sure what it depicts. Is it a petri dish with bacteria colonies? Don’t you have a better photo? Please change it and don’t forget to indicate a source of the photo.

-- Why P, K and Zn are available, and Fe is absorbable. I suggest changing absorbable to available.

-- "excess Na,Cl” is hanging somewhere between arrows , where it belongs ?

-- generally, align text and elements of the figure

-- The figure should “speak for itself” - explain the abbreviations in the legend

- L170-171 “HT-PGPR derived from halophytes”? “ventured among researchers”? Please, rephrase the whole sentence, make it clear and to the point. 

- L183 change to “Hassan et al.” or similar

-  L197 “The pathogenic protective mechanisms …” – I do not understand this sentence, are the pathogens protecting? Who is pathogenic? Please, rephrase.

- L210 change “is” to “are”

- L295 – “cumulations”, use “concentrations”

- L291-298 – this paragraph sounds very general but two of the cited works are about tomatoes and one about beans. Find more works or indicate what plants were the models. How does it relate to rice?

- L365.. – “Ashraf…”  it was shown only in wheat according to the publication.

- L433 – Are the “reduced proline accumulations” one of the “prominent outcomes”? “According to species”? what species? Please rephrase the whole sentence, make it clear and to the point.

- L438-9 – “Additionally,…” – its important to mention – “in some cases” since cross-compatibility was not even tested in the most cited works  

- L456-457 -  “researches are made published” do you mean “researchers published” or something else ?

- L471 – “pandemic”- this word is recently overused for obvious reasons, what you described does not fit to the definition of a pandemic at all- change.

L477 – change to “eco-friendly”

L478 – change “researches” to studies or similar  

Kind regards and good luck with the revision

Author Response

Dear Reviewer, 

I am pleased to inform you that we have carefully revised the manuscript as according to the suggestion given in the attached file, listed point by point.

Regards, 

Ng  

Round 2

Reviewer 3 Report

Dear Authors,

Thank you for addressing the comments. I think the manuscript can be accepted in the present form.

Kind Regards,